# Host–Microbe Interactions and Gut Health in Poultry—Focus on Innate Responses

**DOI:** 10.3390/microorganisms7050139

**Published:** 2019-05-16

**Authors:** Leon J. Broom

**Affiliations:** 1Gut Health Consultancy, Exeter, Devon EX14 1QY, UK; guthealthconsultancy@gmail.com; 2Faculty of Biological Sciences, University of Leeds, Leeds LS2 9JT, UK

**Keywords:** health, heterophil, immunity, immunometabolism, inflammatory response, intestine, macrophage, microorganism, poultry, resolution

## Abstract

Commercial poultry are continually exposed to, frequently pathogenic, microorganisms, usually via mucosal surfaces such as the intestinal mucosa. Thus, understanding host–microbe interactions is vital. Many of these microorganisms may have no or limited contact with the host, while most of those interacting more meaningfully with the host will be dealt with by the innate immune response. Fundamentally, poultry have evolved to have immune responses that are generally appropriate and adequate for their acquired microbiomes, although this is challenged by commercial production practices. Innate immune cells and their functions, encompassing inflammatory responses, create the context for neutralising the stimulus and initiating resolution. Dysregulated inflammatory responses can be detrimental but, being a highly conserved biological process, inflammation is critical for host defence. Heterogeneity and functional plasticity of innate immune cells is underappreciated and offers the potential for (gut) health interventions, perhaps including exogenous opportunities to influence immune cell metabolism and thus function. New approaches could focus on identifying and enhancing decisive but less harmful immune processes, improving the efficiency of innate immune cells (e.g., targeted, efficient microbial killing) and promoting phenotypes that drive resolution of inflammation. Breeding strategies and suitable exogenous interventions offer potential solutions to enhance poultry gut health, performance and welfare.

## 1. Introduction

Commercially raised chickens live in environments with continual exposure to numerous microorganisms, normally via their mucosal surfaces. Traditionally, host–microbe interactions are categorised as mutualistic, commensalistic, or parasitic and more recently, microbes have been classified as symbionts, pathobionts or (true) pathogens [1]. There is, however, a spectrum of possible outcomes arising from ‘contact’ between a host and a microbe. It is proposed that most ‘contacts’ between host and microbe lead to no interaction, while, depending on the microbe and the host response, disease (of varying severity) may result [2].

Many microorganisms, for example those entering the gastrointestinal (GI) tract, may simply be swept through the lumen, along with the digesta by peristalsis, without meaningfully interacting with the host. Any opportunity for such microorganisms to influence the host would be through the production and release of compounds (e.g., short chain fatty acids (SCFA), lipopolysaccharide, etc.) into the intestine. The local (e.g., intestinal) microbiome also competes, for example for nutrients, through production of antimicrobial compounds, occupying colonisation sites, etc., against microorganisms arriving from the external environment. The potential of a more mature gut microbiome to significantly improve resistance to *Salmonella* challenge has been clearly demonstrated in chickens [3]. These initial ‘barriers’ to exogenous microbes are supplemented by host secretions into the gut lumen such as mucus, immunoglobulins (IgA), host defence peptides (HDP) and enzymes (e.g., lysozyme, intestinal alkaline phosphatase, etc.), which seek to trap and eliminate undesirable microorganisms from (whilst helping to maintain commensals within) the GI tract, and thus seek to prevent contact with underlying host cells [4]. In the intestine, the underlying cells separating host tissues from the external environment are a monolayer of epithelial cells linked together by tight junctions (TJ) that regulate paracellular permeability. These cells are essentially differentiated into four major types—goblet cells (mucin production), Paneth cells (HDP secretion), endocrine cells (hormone production) and enterocytes (nutrient absorption). Therefore, various factors contribute to regulating the composition and activity of the (gut) microbiome and the possibility of microbes to infect host tissues and cause disease.

Should microbes overcome these barriers, the host needs further mechanisms to detect, and respond to, their potentially menacing presence. Pattern recognition receptors (PRR) are expressed by numerous host cells, including immune and intestinal epithelial cells (IEC), and are the primary mechanism by which the host can survey microbial activity and respond appropriately. PRR recognise microbe-associated molecular patterns (MAMPs), which are conserved microbial structures, or host-derived danger-associated molecular patterns (DAMPs) arising from cellular damage and the leakage of cytoplasmic and nuclear components. There are various PRR families, which include soluble (such as collectins, complement components, LPS binding protein and pentraxins) and cell-associated (surface or intracellular; such as toll-like receptors (TLR), nucleotide-binding oligomerization domain (NOD)-like receptors (NLR), retinoic acid-inducible gene I (RIG-I)-like receptors (RLR), etc.) components. A thorough review of PRR specific to chickens is provided by Juul-Madsen et al. [5]. Soluble PRR are considered the innate immune system’s equivalent of antibodies and contribute to opsonisation of pathogens and apoptotic cells and regulation of complement activation and inflammation [6]. Engagement of cell-associated PRR by their cognate MAMP or DAMP induces conformational changes, the recruitment of adaptor proteins (e.g., MyD88, TIRAP, TRIF, and TRAM) and the initiation of distinct signalling pathways, culminating in nuclear factor kappa B (NF-κB) and mitogen-activated protein kinase (MAPK) activation [7]. Inactivated, NF-κB is complexed with the inhibitory protein IκBα within the cytosol, but activation of relevant signalling pathway(s) activates the enzyme IκB kinase leading to dissociation and degradation of IκBα, activation of NF-κB and its nuclear translocation, and the production of proinflammatory cytokines and chemokines. Other distinct PRR signalling pathways can lead to activation of IFN regulatory factor (IRF) transcription factors and the production of type I and type III IFNs [8]. Various factors, such as crosstalk between signalling pathways, pathway activation thresholds, feedback loops, etc., can tailor the inflammatory response to the stimulus, while limiting excessive inflammation [9]. Activation of IEC PRR promotes the expression of mucus, IgA, HDP, TJ and regulates cell proliferation/apoptosis, thus supporting a central role for IEC at the interface between the gut microbiota and the host, and in the maintenance of the intestinal barrier and homeostasis [10].

Whilst some immune cells, predominantly γδ T-cells, reside among the epithelial cells (intraepithelial lymphocytes; IEL), the majority are located within the lamina propria below the epithelium. These cells include innate (e.g., dendritic cells (DCs), heterophils (avian equivalent of mammalian neutrophil), macrophages, and natural killer (NK) cells) and adaptive (e.g., T and B cells) immune cells that are distributed throughout the lamina propria or in more organised lymphoid aggregates (e.g., Peyer’s Patches). Macrophages are the first cells to make contact with microbes within infected tissues [11] and macrophage-like cells reportedly have a relative abundance within the intestinal mucosa [12]. Activation of macrophages, via ligation of their PRR, initiates the phagocytosis and destruction (via respiratory burst activity, nitric oxide, etc.) of the detected microbe, and production of various cytokines and chemokines that signal to other components of the immune system, including initiation of adaptive immune responses as necessary/appropriate. These signalling molecules include IL-1β, IL-6, two IL-8-like chemokines (CXCLi1 and CXCLi2), IL-10, IL-18 and TNF-α, with IL-1β, IL-6, IL-18 and chemokine expression seemingly a relatively uniform response of chicken macrophages to microbial stimuli [11]. Differences in specific response patterns have been detected between macrophages from more resistant or susceptible chicken lines and to antigenically distinct strains of a microorganism (e.g., *Salmonella* serovars). CXCLi1 is efficient at recruiting heterophils, which are considered the first responders of the avian innate immune system to infectious insult and injury [13]. In fact, heterophils can mount a rapid response to bacterial infection, within minutes, to phagocytose and kill microorganisms through degranulation (e.g., HDP), oxidative burst and extracellular traps (ET), although heterophils generate a weakened oxidative burst response compared to neutrophils as they lack myeloperoxidase [14]. In response to *Salmonella Enteritidis* (SE) challenge in broilers, nearly all (99%) stained peritoneal leukocyte cells were found to be heterophils [15]. Heterophils express various PRR and have been reported to produce a similar repertoire of cytokines and chemokines (IL-1β, IL-6, IL-8, IL-10, IL-18, TGF-β4 granulocyte macrophage-colony stimulating factor GM-CSF) as chicken macrophages following bacterial challenge (e.g., SE) [14]. Innate immunity is the first line of defence against successful infection and often eliminates invading microorganisms before the host’s adaptive immunity is mobilized [8].

## 2. Inflammation: Innate Response to Noxious Stimuli

The early innate responses to infectious insult and injury include pro-inflammatory mediators. Inflammation is an essential, dynamic, but tightly-regulated immune response that is initiated within seconds of infection and/or tissue injury being detected [16]. Inflammation aims to maintain or restore tissue homeostasis through elimination of the pathogen, removal of damaged tissue and healing processes [17]. Activation of PRR initiates the inflammatory cascade by actively releasing soluble proinflammatory mediators. These mediators, including cytokines (e.g., TNF-α, IFN-γ, IL-1), chemokines (e.g., IL-8), vasoactive amines (e.g., histamine) and eicosanoids (e.g., prostaglandin E2 (PGE2)), work in a coordinated manner to induce vasodilation and vascular permeability, facilitating plasma leakage (including antibodies and complement) and granulocyte influx into the infected tissue [18]. In the simplest model of inflammation, signalling by tissue-resident immune cells causes localised vasodilation and increased blood flow, bringing leukocytes, principally heterophils, into contact with the endothelium. The selectin family of transmembrane adhesion receptors, notably P-selectin, enable the leukocytes to form bonds with the endothelial wall and begin the processes of rolling and adherence via their integrins interacting with endothelial intercellular adhesion molecules (ICAMS) and vascular cell adhesion molecules (VCAMS) [19]. Following adherence, the heterophils translocate through the endothelium via diapedesis and exert strong antibacterial killing mechanisms such as degranulation and oxidative burst (generation of reactive oxygen species (ROS) [20]. This is followed by the recruitment of monocytes to the tissue, and differentiation to macrophages, to clear the heterophils and tissue debris as part of the resolution phase of inflammation. In the first 6–12 h of the acute inflammatory response, heterophil infiltrates dominate, with macrophages and lymphocytes recruited within the next 36 h [21]. Tissue lesions can occur as heterophil granulomas form due to macrophages and fibrous connective tissue walling off necrotic heterophils. Direct tissue damage can also occur from heterophil degranulation [21]. The nature of the stimulus and the effectiveness of the host response will determine the duration of the inflammatory response and subsequent phases (e.g., resolution). Chronic (longer-lasting) inflammation, generally mediated by adaptive immunity (i.e., lymphocytes (and macrophages)), is considered to represent a state by which acute inflammatory mechanisms have failed to eliminate tissue infection and/or injury. It is important to recognise the different phases of the inflammatory response (e.g., acute, persisting and chronic), sites affected (e.g., localised or systemic), mediators (related to innate- vs. adaptive responses) and causes (infectious vs. non-infectious).

Historically, research has focused on the initiation of inflammation with less attention on, and thus understanding of, the resolution of inflammation. Resolution of inflammation is characterised by counter-regulation of proinflammatory mediators, limiting further heterophil influx into the tissue, clearance of heterophils through apoptosis induction and efferocytosis by macrophages or efflux from the tissue, transformation of macrophages to alternatively activated, healing processes and return to homeostasis [20], although now the character of the tissue may not exactly equate to the pre-inflamed state [18]. Previously, it was thought that the resolution of inflammation was a passive process resulting from a decline in proinflammatory mediators and thus the responsiveness of target cells. It is, however, apparent that the resolution of inflammation is an active and regulated process that begins shortly after the onset of the inflammatory response [22]. It is proposed that the resolution of inflammation is initiated by the proinflammatory environment at the site of injury and that mucosal injury simultaneously drives both proinflammatory responses and resolution processes to restore tissue homeostasis [23]. Recent studies have identified a novel group of mediators produced predominantly by innate immune cells that promote clearance of pathogens and activate resolution pathways that restore tissue homeostasis [24]. These mediators have been termed ‘immunoresolvents’ as they reportedly promote the resolution of inflammatory conditions without compromising the ability of the host to counteract invading microorganisms [23]. Pro-resolving mediators are a diverse group, although recent focus has been on the lipid molecules (lipoxins, resolvins, protectins and maresins), which are produced from ω-3 and ω-6 essential fatty acids (Table 1).

## 3. Relevance to Poultry Gut Health and Associated Studies?

### 3.1. Host–Microbe Interactions

Chickens have evolved in environments densely populated with diverse microbes and their ongoing survival confirms the development of suitable immune capability. This would have required host fitness trade-offs to evolve, generally, appropriate immune responses but commercial chickens are now raised in modern, less ‘natural’ environments, potentially undermining suitable evolutionary development. PRR, their signalling pathways and regulation are crucial to the host ‘interpreting’ and responding appropriately to this environmental microbiome. Many microorganisms will transiently come into contact with the chicken host but without meaningful interaction, while others will be tolerated or covertly repelled by the immune system. Microorganisms with greater intent (e.g., virulence) and/or in requisite numbers interact more meaningfully with the host and thus the subsequent progression and outcome (e.g., disease or severity) depends on the interplay between the host and microbe(s).

Macrophages express various PRR and, due to their location and number, detect invasive microorganisms and contribute significantly to the course of infection, not least because they themselves are susceptible to pathogen infection. Although ligation of PRR is most often thought of as leading to the production of proinflammatory mediators, potentially there are various outcomes as numerous negative regulators of, for example, TLR signalling are known, including TLR degradation or suppression, intracellular inhibitors, down-regulation of gene transcription or post-transcriptional regulation by microRNAs (miRNAs) [25]. The importance of negative regulators of NF-κB signalling in controlling cellular responses have been demonstrated in knock-out models [9]. Moreover, the specific structure of the PRR ligand appears to influence the host cell response and oral feeding of TLR ligands (e.g., lipopolysaccharide and lipoteichoic acid) can correct perturbed signalling (e.g., colitis), supporting the concept of an active role for ligand recognition in homeostasis [25]. Similarly, a common strategy employed by pathogens is to target PRR pathways via virulence factors, for example with proteolytic activity against MyD88 and TRIF, to manipulate cellular signalling [9]. However, it can be anticipated that detection of microbes within host tissues by macrophages typically induces a positive, proinflammatory response to rapidly recruit and activate innate immune cells and prime adaptive immune responses. In addition, more recent evidence/hypotheses suggest that the proinflammatory environment also drives the initiation of resolution processes [23]. Macrophages are recognised for their recruitment of heterophils to the site of infection, notably through chemokine (e.g., CXCLi1 (IL-8-like)) secretion.

Heterophil/neutrophil recruitment to the infected site is a key feature of the inflammatory response and is regarded as a hallmark of inflammation. Neutrophils have traditionally been considered as relatively limited, one dimensional immune cells but recently their broader role in immune responses has been recognised and actively investigated. These studies have highlighted the longer lifespan of neutrophils than previously thought, their phenotypic plasticity, role in the resolution of inflammation and modulation of an adaptive response [26]. Although neutrophils have unusually high levels of nuclear IκBα to inhibit NF-κB-mediated gene expression and undergo apoptosis promptly following activation, their potential to contribute to tissue damage has been recognised for some time [9]. However, the contribution of heterophils and neutrophils to the resolution of inflammation is less well understood. While neutrophil recruitment is often considered a double-edged sword, in neutrophil depletion models, wound healing is delayed and autoimmune diseases can be exacerbated, indicating that neutrophils play a key role in the resolution of inflammation and/or tissue repair, and that sufficient numbers are necessary for these processes [20]. In addition, aggregated neutrophil ET degrade inflammatory cytokines [27] and thus contribute to the counter-regulation of inflammatory mediators, which is a characteristic of resolution. Accumulating evidence, therefore, indicates that recruited neutrophils are crucial both as primary responders to infection through the inflammatory process and in contributing to the resolution of inflammation and mucosal repair. In fact, neutrophils entering infected tissues during acute inflammation are now considered important for promoting a lipid mediator class switch from the proinflammatory eicosanoids (e.g., prostaglandins and leukotrienes) to pro-resolving mediators (e.g., lipoxins), with proinflammatory cytokines reportedly upregulating enzymes responsible for the synthesis of pro-resolving mediators and expression of their receptors [20]. Whilst some of these aspects will need verifying in chickens, it is likely that the functions of heterophils are far more nuanced than is currently appreciated. Additionally, the role of inflammatory mediators can be nuanced. For example, prostaglandins, notably PGE2, support gastrointestinal homeostasis and epithelial cell health, and their inhibition (by indomethacin) disrupted the immune response, IEC TJ and the gut microbiota in a mouse model of Clostridium difficile infection [28].

The importance of heterophils and their functioning to chicken health has been demonstrated by various studies [14]. Depletion of heterophils by 5-fluorouracil resulted in SE gastrointestinal infection disseminating systemically in young chickens [29] and circulating heterophil numbers have been positively correlated with resistance to organ invasion by SE [30]. Notably, heterophil depletion negatively affected bodyweight and mortality, although non-depleted birds had significantly more intestinal lesions (but less severe extraintestinal infection) [29]. Recruitment, activation and/or functional capability of heterophils can be enhanced through T-cell-derived cytokines from SE-immune chicks [31] or flagellin recognition by TLR5, which limits flagellated Salmonella translocation from the intestine [32]. It has been proposed that enhancing heterophil functions of commercial chickens and turkeys may improve resistance to bacterial infections in young poultry and enhance lifetime health [14]. Neutrophils are also considered to have key roles in viral [33], parasitic [34] and fungal [35] infections but work is lacking in these areas regarding heterophils. 

### 3.2. Inflammatory Responses

Recent reviews have highlighted the potential negative consequences of inflammatory processes in chickens e.g., [36] but studies comparing more susceptible with more resistant birds, for a range of important poultry pathogens, suggest that innate immune cells from more resistant birds respond more rapidly, show enhanced functional characteristics and support a more inflammatory environment in relevant tissues [37]. It is also worth noting that *Campylobacter jejuni* infection in chickens was reported to induce proinflammatory chemokine expression and influx of heterophils into gut tissues, but without any pathology [38]. Similarly, Foster et al. [39] showed that *Salmonella* strains can initiate polymorphonuclear infiltration into the intestinal tissue of gnotobiotic pigs but without pathology. Moreover, neutrophil migration into the lung without accompanying activation does not cause tissue injury [40]. Therefore, heterophil/neutrophil recruitment per se cannot be used as a proxy for tissue pathology as the relationship between the two is not clear-cut. As mentioned above, with SE challenge, maintaining heterophil numbers was associated with greater intestinal lesions but also higher bodyweight [29]. Likewise, where available, better growth performance is a parameter used to define birds more resistant to common poultry pathogens and whose tissues are generally supporting a more inflammatory milieu and/or responsive immune cells [37]. While it seems obvious that maintaining the integrity of the intestine should correlate with better nutrient absorption and thus growth, various studies do not show clear relationships between gut damage (e.g., lesion scores) and chicken growth performance e.g., [41,42,43].

IL-10 is a key immunoregulatory cytokine that functions to downregulate inflammation. There are advocates for ‘anti-inflammatory’ strategies, focusing on the upregulation of IL-10 and/or downregulation of ‘pro-inflammatory’ cytokines in response to stimuli. IL-10 has, however, been recognised to be immunosuppressive and we recently found that it was reportedly more upregulated in relevant tissues following infection with key poultry pathogens in more susceptible vs. more resistant chickens [37]. IL-10 can be expressed by various chicken immune cells [44]. IL-10 was not detected in the serum of healthy, uninfected chickens but was substantially increased in serum 5 days post-infection (dpi) of birds challenged with both a high and low dose of *E. tenella*, suggesting that serum IL-10 could be used as a marker of infection [44]. Recently, a significant positive correlation was found between caecal lesion score and serum IL-10, with both traits negatively correlated with bodyweight gain [45]. In this study, serum IL-10 was used as a measure of the innate immune response to *Eimeria* infection and was considered to reflect the degree of intestinal pathology. It is certainly appreciated that serum cytokine levels can reflect intestinal levels/changes [46]. However, the relationship described by Boulton et al. [45] could also be interpreted as IL-10 contributing to tissue pathology through immunosuppression and worsening disease. IL-10 has been shown to be constitutively expressed (e.g., splenocytes), as well as in response to infection [47]. Moreover, feeding oral antibodies against IL-10 improves the growth performance of *Eimeria* challenged broiler chickens, presumably through neutralisation of IL-10 in the gut lumen, thus preventing binding to the apical IL-10 receptor on enterocytes and reducing IL-10 influence (e.g., through inhibition of IFN-γ production) in the mucosa [48,49]. Whilst a balance is required between excessive immunoregulation and excessive tissue damage in response to (intracellular) microbial infection, the critical point maybe the degree of relative (to proinflammatory mediators) upregulation of immunoregulatory cytokines (e.g., IL-10) post infection [37]. With this in mind, it could be of interest to consider exploring cytokine ratios (e.g., IL-10: IFN-γ) in future studies. Additionally, timing and/or duration of (relative) upregulation are likely to be relevant. For example, TNF-α aids clearance of respiratory syncytial virus (RSV) during the early phase of infection but prolonged production is suggested to exacerbate tissue damage [50]. Changes in pro- or anti-inflammatory biomarkers may not, however, reflect changes in functional immunity and thus directly assessing cellular function may be necessary [51]. Serum IL-10 seems a potential marker for intestinal pathology, notably relating to *Eimeria* infection, but strategies that seek to upregulate immunoregulatory or anti-inflammatory, or downregulate proinflammatory, mediators, particularly in the very early phase of infection, maybe ill-founded, particularly given the majority of important poultry pathogens are intracellular.

Perspectives on inflammatory responses maybe skewed by chronic inflammatory conditions in humans. Acute inflammation is often self-limiting and normally resolves once the stimuli are eliminated and homeostasis is then restored. Overzealous, unresolved or prolonged inflammation becomes an issue and may result in chronic disorders or autoimmune diseases [18]. Some very virulent viral strains (e.g., of infectious bursal disease virus) may cause greater immune system dysregulation, indicated by a stronger proinflammatory response, and pathology e.g., [52] but the very early immune dynamics (<1 dpi) are typically not studied. Effective innate immunity, including acute inflammatory responses, are, however, critical in preventing the successful establishment of infection and can eliminate the threat before mobilisation of adaptive immunity. If established, an appropriate adaptive immune response can take as long as 1 to 2 weeks to clear an infection [53]. Shortening the duration of immune system activation could limit the degree of systemic involvement (e.g., hepatic acute phase response) and associated consequences (e.g., nutrient diversion, anorexia, etc.,). Moreover, ongoing/prolonged infection risks fueling a vicious circle of PRR activation through the release of DAMP. Humans with genetic deficiencies in important components of the inflammatory process have increased risk of infections [54]. Treatment of some chronic inflammatory disorders in humans (e.g., inflammatory bowel disease maybe attempted through the use of steroids that have potent anti-inflammatory actions or binding prominent cytokines (e.g., TNF), but such strategies are reported to increase the incidence of infections [55] and, paradoxically, decrease regulatory lymphocyte activity [56]. In addition, many molecules developed for ‘anti-inflammatory’ application in humans never came to market or are ineffective in a significant proportion of patients [57]. Interestingly, primary immunodeficiencies (e.g., AIDS) in humans are commonly associated with intestinal disorders [56]. It is debatable whether comparisons should be made between treatment of chronic inflammatory conditions in humans living in relatively sanitary conditions and commercially raised animals in less hygienic environments and invariably experiencing significant pathogen challenge. However, there are various (e.g., non-microbial) initiators of inflammatory responses in animals (and humans) and not all of these represent acute challenge to the host and should be mitigated [58].

Sepsis, a life-threatening condition of organ dysfunction caused by a dysregulated immune response to infection [59], is a leading cause of death in humans and animals and may exemplify the complexity of understanding immune responses. Superficially, sepsis may simply be considered the result of the overwhelming release of pro-inflammatory cytokines (“cytokine storm”). However, it is the failure of the early innate immune response to effectively control and clear the pathogen that leads to dysregulation and the “cytokine storm” [60]. This may include impaired neutrophil migration to sites of infection during this initial phase [61]. The excessive inflammatory response is counteracted by anti-inflammatory mediators inducing prolonged immunosuppression or “immunoparalysis”, which is suggested to play a key role in the pathogenesis and disease outcome [62]. The young and old, due to immaturity or compromised functioning, respectively, are predisposed to sepsis. Therapeutic options for sepsis, including inhibiting inflammatory cytokines, have been unsuccessful and attention has now turned to alternative approaches, such as targeting the metabolism of immune cells (immunometabolism) as this influences their functional state, which has been shown to change during sepsis [60]. Similarly, it has recently been reported that butyrate can shift macrophage metabolism, enhancing antimicrobial function and resistance to enteropathogens [63]. Likewise, SCFA generated by the gut microbiome may be absorbed and transported to the bone marrow where they can influence the local environment, influencing haematopoiesis and the types of immune cells generated, and thus immune responses at various sites [64]. Thus, immunometabolism could be extremely relevant for both existing and future strategies to improve poultry gut health [65] and there are likely to be learnings from conditions such as sepsis that can help inform our approaches, which may include promoting early innate responses to effectively control pathogens and lessen their downstream effects.

## 4. Conclusions

Modern poultry, including immune responses, have co-evolved with their natural environmental microbiomes and thus the survival of poultry species implies appropriate immune system development/responses. However, commercial poultry are now raised in environments that are appreciably different to those of their evolving ancestors and thus understanding host–microbe interactions is of critical importance. Microbes are principally detected by PRR expressed by various host cells and the cell’s response is influenced by ligand number, composition and dose, the PRR engaged and cellular location, crosstalk between pathways, activation frequency and threshold(s) and regulation/feedback loops. Thus, the cell’s response is typically stimulus-specific and appropriate, and, when necessary, involves signalling to, and recruiting, other cells such as heterophils. Work in humans and mice is establishing that the traditional view of neutrophils needs reviewing as these heterogenous cells have diverse functions, including involvement in adaptive responses and resolution of inflammation. It can be predicted that heterophils have greater phenotypic and functional diversity than is currently appreciated. These insights definitely complicate binary perspectives of heterophils and therapeutic strategies (e.g., blocking heterophils) arising from them. Heterophils and optimal functioning appear to be important for host protection, particularly in early life when adaptive immunity is immature and/or has delayed responses.

There is an understandable desire to identify biomarkers that can be used to reflect host status and responses to interventions. However, given the complexity of immune signalling pathways, cellular interactions and context-specific responses, it is important that care is exercised when interpreting measured experimental parameters and developing intervention strategies arising from them. Individual or small numbers of parameters are likely to be misleading and a panel of suitable analyses, at appropriate timepoints, are necessary.

Inflammation is one of the most highly conserved biological processes, suggesting its critical importance for host defence [66]. A pro-inflammatory environment facilitates rapid control, and likely elimination, of the infection, as well as resolution of the inflammatory response. As outlined, inappropriate modulation of these responses may have detrimental, and possibly life-threatening, consequences. Thus, there is growing recognition that popular ‘anti-inflammatory’ strategies should be superseded by ‘immunoresolving’ approaches that resolve inflammation without compromising host defences. More work is required to validate the role and significance of pro-resolving mediators in poultry [67], but various mouse studies with, for example, supplemental ω-3 and ω-6 essential fatty acids have been effective [20] and this may prove a fruitful area for academic and commercial research. Innate immune cells recruited in the early phase of infectious insult and injury maybe decisive in creating the necessary inflammatory context to initiate the resolution phase and production of pro-resolving mediators. Overzealous or non-resolving inflammatory responses are not desirable and maybe reflective of a suboptimal early innate response. Fully understanding the causes of dysregulated inflammatory responses and failure to resolve will lead to appropriate interventions in the future.

There are various approaches for modulating gut health in poultry (Table 2). These strategies primarily focus on exogenously influencing the intestinal microbiome and/or host response(s) but could also include selecting poultry that are resistant (resist infection), tolerant (tolerate the consequences of infection), or resilient (recover from consequences of infection promptly) to microbial infection [45]. As well as disease susceptibility differences between different bird lines, innate resistance seems to differ between sexes (e.g., to *E. tenella*) [68] and the underlying mechanisms should be explored further. Exogenous approaches could include the use of PRR ligands (e.g., β-glucans, CpG ODN, etc.) [69,70], or other suitable additives, to prime innate immune cells. Improving the efficacy of tissue resident immune cells could reduce the need to escalate the immune response following infection and/or injury. Less superficially, delineating protective immune mechanisms from less critical but more harmful responses will be important and data are showing that this seems possible. For example, SCFA can influence haematopoiesis and/or immune cell metabolism and function, while more recently appreciated innate immune memory (differing subsequent innate immune cell responses to previous stimulus) may also offer opportunities. Moreover, commonly used feed additives (e.g., probiotics, prebiotics, phytogenics, organic acids, enzymes, etc.) can help positively influence host-microbiome interactions to support efficient growth and health. However, we should also keep in mind that well-intentioned interventions could potentially disrupt intestinal homeostasis [56].

This article outlines some of the complexities and nuances of host immune responses and their interpretation. We need to better understand host–microbe interactions to enable more precise application of suitable interventions.

## Figures and Tables

**Table 1 microorganisms-07-00139-t001:** Pro-resolving mediators and their precursors.

Polyunsaturated Fatty Acid Precursors	Pro-Resolving Family/Series
Arachidonic acid (C20:4)	Lipoxin
Eicosapentanoic acid (C20:5)	Resolvin E
Docosapentaenoic acid (C22:5)	Resolvin D, E & T, Maresin, Protectin
Docosahexanoic acid (C22:6)	Resolvin D, Maresin, Protectin

**Table 2 microorganisms-07-00139-t002:** Potential strategies to promote poultry gut health.

Strategy	Example(s)	Desired Outcome
**Breeding**		
	↑ Resistance	Resist infection (and/or)
	↑ Tolerance	Tolerate consequences (and/or)
	↑ Resilience	Recover promptly
**Exogenous**		
Prime innate immune cells	PRR ligands	Potentiate immune cells
Immunometabolism	SCFA	Influence immune cell phenotype & function
Resolution	ω-3 and ω-6 essential fatty acids	Promote production of pro-resolving mediators
Microbiome-host interactions	Probiotics Prebiotics Phytogenics Organic acids Enzymes	Influence gut microbiome function and/or host responses

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
