# Peer review of "Host–Microbe Interactions and Gut Health in Poultry—Focus on Innate Responses"

_microorganisms, 2019, doi:10.3390/microorganisms7050139_

Reviewer 1 Report

This is a generally excellent and very readable review on the topic of host-microbe interactions and poultry gut health. The author makes the topic, including the immunology, understandable. This paper should become a good resource for non-experts looking for a well-rounded introduction to the field.

Minor Comments:

Lines 46-49: It may be of value to include a brief discussion of intestinal alkaline phosphatase in this section. It is a significant secreted enzyme in host-microbe interactions.

Lines 244-246: A discussion of the work of Kirk Klasing and Doug Korver may be relevant here. They have done extensive work on how the inflammatory response effects energy balance and growth. Their results appear at odds with this description.

Lines 308-321: There should be a clear link drawn between sepsis, specifically, and chicken gut health. Are there references for sepsis or bacterial invasion of blood across the gut barrier and cytokine storm, in chicken?

Author Response

Dear Reviewer,

Thank you for your positive and helpful review.

Lines 46-49: Intestinal alkaline phosphatase has been included in this section.

Lines 244-246: The reader is referred to an example review reference [36] at the beginning of the paragraph that discusses the negative consequences/nutritional impact of inflammatory responses, including work by Kirk Klasing, whereas lines 244-246 are specifically referring to challenge models using birds defined as more resistant or susceptible to the particular poultry pathogen. The review seeks to promote the concept of balanced immune responses but there is significant evidence that early control of infection through greater, including inflammatory mediator, responsiveness reduces dysregulated or prolonged responses and associated nutritional and growth consequences. 

Lines 308-321: I could not find directly relevant references. However, the intention of the sepsis-related paragraph is to highlight shifts in the perception of a dysregulated immune response (e.g. sepsis) from simply an overzealous proinflammatory immune response to failure (and thus the importance) of the early innate immune response to control infection, that immature or suboptimal immune functioning (e.g. young or old birds) predisposes, and the potential of immunometabolism to shape the response, all of which are relevant to poultry gut health management. 

Kind regards

Leon   

Reviewer 2 Report

The manuscript was properly conducted and findings reported are important for poultry production and health. The paper contains important data health of chickens under different dietary treatments. The Authors investigated an interesting topic and the objective of the paper is of worldwide interest and fits well within the overall scope of the journal. Results were properly reported and the findings have been accurately discussed and compared with other recently published papers.

Author Response

Dear Reviewer,

Thank you for your positive review and comments.

Best wishes

Leon

Reviewer 3 Report

Well written review.  Very thorough.  Of importance to the poultry industry, especially related to broilers and cage-free laying hens and their gut health.  

Author Response

Dear Reviewer,

Many thanks for your review and encouraging comments.

Best regards

Leon